# STRUCTURED ATTENTION NETWORKS

**Yoon Kim**\*    **Carl Denton**\*    **Luong Hoang**    **Alexander M. Rush**
{yoonkim@seas, carldenton@college, lhoang@g, srush@seas}.harvard.edu
School of Engineering and Applied Sciences
Harvard University
Cambridge, MA 02138, USA

## ABSTRACT

Attention networks have proven to be an effective approach for embedding categorical inference within a deep neural network. However, for many tasks we may want to model richer structural dependencies without abandoning end-to-end training. In this work, we experiment with incorporating richer structural distributions, encoded using graphical models, within deep networks. We show that these structured attention networks are simple extensions of the basic attention procedure, and that they allow for extending attention beyond the standard soft-selection approach, such as attending to partial segmentations or to subtrees. We experiment with two different classes of structured attention networks: a linear-chain conditional random field and a graph-based parsing model, and describe how these models can be practically implemented as neural network layers. Experiments show that this approach is effective for incorporating structural biases, and structured attention networks outperform baseline attention models on a variety of synthetic and real tasks: tree transduction, neural machine translation, question answering, and natural language inference. We further find that models trained in this way learn interesting unsupervised hidden representations that generalize simple attention.

## 1 INTRODUCTION

Attention networks are now a standard part of the deep learning toolkit, contributing to impressive results in neural machine translation (Bahdanau et al., 2015; Luong et al., 2015), image captioning (Xu et al., 2015), speech recognition (Chorowski et al., 2015; Chan et al., 2015), question answering (Hermann et al., 2015; Sukhbaatar et al., 2015), and algorithm-learning (Graves et al., 2014; Vinyals et al., 2015), among many other applications (see Cho et al. (2015) for a comprehensive review). This approach alleviates the bottleneck of compressing a source into a fixed-dimensional vector by equipping a model with variable-length memory (Weston et al., 2014; Graves et al., 2014; 2016), thereby providing random access into the source as needed. Attention is implemented as a hidden layer which computes a categorical distribution (or hierarchy of categorical distributions) to make a soft-selection over source elements.

Noting the empirical effectiveness of attention networks, we also observe that the standard attention-based architecture does not directly model any *structural dependencies* that may exist among the source elements, and instead relies completely on the hidden layers of the network. While one might argue that these structural dependencies can be learned implicitly by a deep model with enough data, in practice, it may be useful to provide a structural bias. Modeling structural dependencies at the final, *output* layer has been shown to be important in many deep learning applications, most notably in seminal work on graph transformers (LeCun et al., 1998), key work on NLP (Collobert et al., 2011), and in many other areas (Peng et al., 2009; Do & Artiéres, 2010; Jaderberg et al., 2014; Chen et al., 2015; Durrett & Klein, 2015; Lample et al., 2016, *inter alia*).

In this work, we consider applications which may require structural dependencies at the attention layer, and develop *internal* structured layers for modeling these directly. This approach generalizes categorical soft-selection attention layers by specifying possible structural dependencies in a soft

---
\*Equal contribution.

manner. Key applications will be the development of an attention function that segments the source input into subsequences and one that takes into account the latent recursive structure (i.e. parse tree) of a source sentence.

Our approach views the attention mechanism as a graphical model over a set of latent variables. The standard attention network can be seen as an expectation of an annotation function with respect to a single latent variable whose categorical distribution is parameterized to be a function of the source. In the general case we can specify a graphical model over multiple latent variables whose edges encode the desired structure. Computing forward attention requires performing inference to obtain the expectation of the annotation function, i.e. the *context vector*. This expectation is computed over an exponentially-sized set of structures (through the machinery of graphical models/structured prediction), hence the name *structured attention* network. Notably each step of this process (including inference) is differentiable, so the model can be trained end-to-end without having to resort to deep policy gradient methods (Schulman et al., 2015).

The differentiability of inference algorithms over graphical models has previously been noted by various researchers (Li & Eisner, 2009; Domke, 2011; Stoyanov et al., 2011; Stoyanov & Eisner, 2012; Gormley et al., 2015), primarily outside the area of deep learning. For example, Gormley et al. (2015) treat an entire graphical model as a differentiable circuit and backpropagate risk through variational inference (loopy belief propagation) for minimium risk training of dependency parsers. Our contribution is to combine these ideas to produce structured *internal* attention layers within deep networks, noting that these approaches allow us to use the resulting marginals to create new features, as long as we do so a differentiable way.

We focus on two classes of structured attention: linear-chain conditional random fields (CRFs) (Lafferty et al., 2001) and first-order graph-based dependency parsers (Eisner, 1996). The initial work of Bahdanau et al. (2015) was particularly interesting in the context of machine translation, as the model was able to implicitly learn an *alignment model as a hidden layer*, effectively embedding inference into a neural network. In similar vein, under our framework the model has the capacity to learn a *segmenter as a hidden layer* or a *parser as a hidden layer*, without ever having to see a segmented sentence or a parse tree. Our experiments apply this approach to a difficult synthetic reordering task, as well as to machine translation, question answering, and natural language inference. We find that models trained with structured attention outperform standard attention models. Analysis of learned representations further reveal that interesting structures emerge as an internal layer of the model. All code is available at `http://github.com/harvardnlp/struct-attn`.

## 2 BACKGROUND: ATTENTION NETWORKS

A standard neural network consist of a series of non-linear transformation layers, where each layer produces a fixed-dimensional hidden representation. For tasks with large input spaces, this paradigm makes it hard to control the interaction between components. For example in machine translation, the source consists of an entire sentence, and the output is a prediction for each word in the translated sentence. Utilizing a standard network leads to an information bottleneck, where one hidden layer must encode the entire source sentence. Attention provides an alternative approach.[1] An attention network maintains a set of hidden representations that scale with the size of the source. The model uses an internal inference step to perform a soft-selection over these representations. This method allows the model to maintain a variable-length memory and has shown to be crucially important for scaling systems for many tasks.

Formally, let $x = [x_1, \ldots, x_n]$ represent a sequence of inputs, let $q$ be a query, and let $z$ be a categorical latent variable with sample space $\{1, \ldots, n\}$ that encodes the desired selection among these inputs. Our aim is to produce a *context* $c$ based on the sequence and the query. To do so, we assume access to an *attention distribution* $z \sim p(z \mid x, q)$, where we condition $p$ on the inputs $x$ and a query $q$. The *context* over a sequence is defined as expectation, $c = \mathbb{E}_{z \sim p(z \mid x, q)}[f(x, z)]$ where $f(x, z)$ is an *annotation function*. Attention of this form can be applied over any type of input, however, we will primarily be concerned with "deep" networks, where both the annotation function

---

[1]Another line of work involves marginalizing over latent variables (e.g. latent alignments) for sequence-to-sequence transduction (Kong et al., 2016; Lu et al., 2016; Yu et al., 2016; 2017).

and attention distribution are parameterized with neural networks, and the context produced is a vector fed to a downstream network.

For example, consider the case of attention-based neural machine translation (Bahdanau et al., 2015). Here the sequence of inputs $[\mathbf{x}_1, \ldots, \mathbf{x}_n]$ are the hidden states of a recurrent neural network (RNN), running over the words in the source sentence, $\mathbf{q}$ is the RNN hidden state of the target decoder (i.e. vector representation of the query $q$), and $z$ represents the source position to be attended to for translation. The attention distribution $p$ is simply $p(z = i \mid x, q) = \text{softmax}(\theta_i)$ where $\theta \in \mathbb{R}^n$ is a parameterized potential typically based on a neural network, e.g. $\theta_i = \text{MLP}([\mathbf{x}_i; \mathbf{q}])$. The annotation function is defined to simply return the selected hidden state, $f(\mathbf{x}, z) = \mathbf{x}_z$. The context vector can then be computed using a simple sum,

$$\mathbf{c} = \mathbb{E}_{z \sim p(z \mid x, q)}[f(x, z)] = \sum_{i=1}^{n} p(z = i \mid x, q)\mathbf{x}_i \tag{1}$$

Other tasks such as question answering use attention in a similar manner, for instance by replacing source $[x_1, \ldots, x_n]$ with a set of potential facts and $q$ with a representation of the question.

In summary we interpret the attention mechanism as taking the expectation of an annotation function $f(x, z)$ with respect to a latent variable $z \sim p$, where $p$ is parameterized to be function of $x$ and $q$.

# 3 STRUCTURED ATTENTION

Attention networks simulate selection from a set using a soft model. In this work we consider generalizing selection to types of attention, such as selecting chunks, segmenting inputs, or even attending to latent subtrees. One interpretation of this attention is as using soft-selection that considers all possible structures over the input, of which there may be exponentially many possibilities. Of course, this expectation can no longer be computed using a simple sum, and we need to incorporate the machinery of inference directly into our neural network.

Define a structured attention model as being an attention model where $z$ is now a vector of discrete latent variables $[z_1, \ldots, z_m]$ and the attention distribution is $p(z \mid x, q)$ is defined as a *conditional random field* (CRF), specifying the independence structure of the $z$ variables. Formally, we assume an undirected graph structure with $m$ vertices. The CRF is parameterized with clique (log-)potentials $\theta_C(z_C) \in \mathbb{R}$, where the $z_C$ indicates the subset of $z$ given by clique $C$. Under this definition, the attention probability is defined as, $p(z \mid x, q; \theta) = \text{softmax}(\sum_C \theta_C(z_C))$, where for symmetry we use softmax in a general sense, i.e. $\text{softmax}(g(z)) = \frac{1}{Z} \exp(g(z))$ where $Z = \sum_{z'} \exp(g(z'))$ is the implied partition function. In practice we use a neural CRF, where $\theta$ comes from a deep model over $x, q$.

In structured attention, we also assume that the annotation function $f$ factors (at least) into clique annotation functions $f(x, z) = \sum_C f_C(x, z_C)$. Under standard conditions on the conditional independence structure, inference techniques from graphical models can be used to compute the forward-pass expectations and the context:

$$c = \mathbb{E}_{z \sim p(z \mid x, q)}[f(x, z)] = \sum_C \mathbb{E}_{z \sim p(z_C \mid x, q)}[f_C(x, z_C)]$$

## 3.1 EXAMPLE 1: SUBSEQUENCE SELECTION

Suppose instead of soft-selecting a single input, we wanted to explicitly model the selection of contiguous subsequences. We could naively apply categorical attention over all subsequences, or hope the model learns a multi-modal distribution to combine neighboring words. Structured attention provides an alternate approach.

Concretely, let $m = n$, define $z$ to be a random vector $z = [z_1, \ldots, z_n]$ with $z_i \in \{0, 1\}$, and define our annotation function to be, $f(x, z) = \sum_{i=1}^{n} f_i(x, z_i)$ where $f_i(x, z_i) = \mathbb{1}\{z_i = 1\}\mathbf{x}_i$. The explicit expectation is then,

$$\mathbb{E}_{z_1, \ldots, z_n}[f(x, z)] = \sum_{i=1}^{n} p(z_i = 1 \mid x, q)\mathbf{x}_i \tag{2}$$

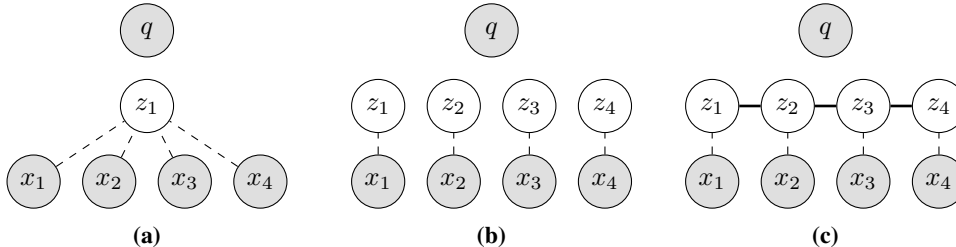

**Figure 1:** Three versions of a latent variable attention model: (a) A standard soft-selection attention network, (b) A Bernoulli (sigmoid) attention network, (c) A linear-chain structured attention model for segmentation. The input and query are denoted with $x$ and $q$ respectively.

Equation (2) is similar to equation (1)—both are a linear combination of the input representations where the scalar is between $[0, 1]$ and represents how much attention should be focused on each input. However, (2) is fundamentally different in two ways: (i) it allows for multiple inputs (or no inputs) to be selected for a given query; (ii) we can incorporate structural dependencies across the $z_i$'s. For instance, we can model the distribution over $z$ with a linear-chain CRF with pairwise edges,

$$p(z_1, \dots, z_n \,|\, x, q) = \text{softmax} \left( \sum_{i=1}^{n-1} \theta_{i,i+1}(z_i, z_{i+1}) \right) \tag{3}$$

where $\theta_{k,l}$ is the pairwise potential for $z_i = k$ and $z_{i+1} = l$. This model is shown in Figure 1c. Compare this model to the standard attention in Figure 1a, or to a simple Bernoulli (sigmoid) selection method, $p(z_i = 1 \,|\, x, q) = \text{sigmoid}(\theta_i)$, shown in Figure 1b. All three of these methods can use potentials from the same neural network or RNN that takes $x$ and $q$ as inputs.

In the case of the linear-chain CRF in (3), the marginal distribution $p(z_i = 1 \,|\, x)$ can be calculated efficiently in linear-time for all $i$ using message-passing, i.e. the forward-backward algorithm. These marginals allow us to calculate (2), and in doing so we implicitly sum over an exponentially-sized set of structures (i.e. all binary sequences of length $n$) through dynamic programming. We refer to this type of attention layer as a *segmentation attention* layer.

Note that the forward-backward algorithm is being used as parameterized *pooling* (as opposed to output computation), and can be thought of as generalizing the standard attention softmax. Crucially this generalization from vector softmax to forward-backward is just a series of differentiable steps,[2] and we can compute gradients of its output (marginals) with respect to its input (potentials). This will allow the structured attention model to be trained end-to-end as part of a deep model.

### 3.2 EXAMPLE 2: SYNTACTIC TREE SELECTION

This same approach can be used for more involved structural dependencies. One popular structure for natural language tasks is a dependency tree, which enforces a structural bias on the recursive dependencies common in many languages. In particular a dependency tree enforces that each word in a source sentence is assigned exactly one parent word (*head word*), and that these assignments do not cross (projective structure). Employing this bias encourages the system to make a soft-selection based on learned syntactic dependencies, without requiring linguistic annotations or a pipelined decision.

A dependency parser can be partially formalized as a graphical model with the following cliques (Smith & Eisner, 2008): latent variables $z_{ij} \in \{0, 1\}$ for all $i \neq j$, which indicates that the $i$-th word is the parent of the $j$-th word (i.e. $x_i \rightarrow x_j$); and a special global constraint that rules out configurations of $z_{ij}$'s that violate parsing constraints (e.g. one head, projectivity).

The parameters to the graph-based CRF dependency parser are the potentials $\theta_{ij}$, which reflect the score of selecting $x_i$ as the parent of $x_j$. The probability of a parse tree $z$ given the sentence

---

[2]As are other dynamic programming algorithms for inference in graphical models, such as (loopy and non-loopy) belief propagation.

**procedure** FORWARDBACKWARD($\theta$)
 $\alpha[0, \langle t \rangle] \leftarrow 0$
 $\beta[n + 1, \langle t \rangle] \leftarrow 0$
 **for** $i = 1, \ldots, n; c \in \mathcal{C}$ **do**
 $\alpha[i, c] \leftarrow \bigoplus_y \alpha[i - 1, y] \otimes \theta_{i-1,i}(y, c)$
 **for** $i = n, \ldots, 1; c \in \mathcal{C}$ **do**
 $\beta[i, c] \leftarrow \bigoplus_y \beta[i + 1, y] \otimes \theta_{i,i+1}(c, y)$
 $A \leftarrow \alpha[n + 1, \langle t \rangle]$
 **for** $i = 1, \ldots, n; c \in \mathcal{C}$ **do**
 $p(z_i = c \,|\, x) \leftarrow \exp(\alpha[i, c] \otimes \beta[i, c]$
 $\otimes - A)$
 **return** $p$

**procedure** BACKPROPFORWARDBACKWARD($\theta, p, \nabla_p^{\mathcal{L}}$)
 $\nabla_\alpha^{\mathcal{L}} \leftarrow \log p \otimes \log \nabla_p^{\mathcal{L}} \otimes \beta \otimes -A$
 $\nabla_\beta^{\mathcal{L}} \leftarrow \log p \otimes \log \nabla_p^{\mathcal{L}} \otimes \alpha \otimes -A$
 $\hat{\alpha}[0, \langle t \rangle] \leftarrow 0$
 $\hat{\beta}[n + 1, \langle t \rangle] \leftarrow 0$
 **for** $i = n, \ldots 1; c \in \mathcal{C}$ **do**
 $\hat{\beta}[i, c] \leftarrow \nabla_\alpha^{\mathcal{L}}[i, c] \oplus \bigoplus_y \theta_{i,i+1}(c, y) \otimes \hat{\beta}[i + 1, y]$
 **for** $i = 1, \ldots, n; c \in \mathcal{C}$ **do**
 $\hat{\alpha}[i, c] \leftarrow \nabla_\beta^{\mathcal{L}}[i, c] \oplus \bigoplus_y \theta_{i-1,i}(y, c) \otimes \hat{\alpha}[i - 1, y]$
 **for** $i = 1, \ldots, n; y, c \in \mathcal{C}$ **do**
 $\nabla_{\theta_{i-1,i}(y,c)}^{\mathcal{L}} \leftarrow \text{signexp}(\hat{\alpha}[i, y] \otimes \beta[i + 1, c]$
 $\oplus \alpha[i, y] \otimes \hat{\beta}[i + 1, c]$
 $\oplus \alpha[i, y] \otimes \beta[i + 1, c] \otimes -A)$
 **return** $\nabla_\theta^{\mathcal{L}}$

**Figure 2:** Algorithms for linear-chain CRF: (left) computation of forward-backward tables $\alpha$, $\beta$, and marginal probabilities $p$ from potentials $\theta$ (forward-backward algorithm); (right) backpropagation of loss gradients with respect to the marginals $\nabla_p^{\mathcal{L}}$. $\mathcal{C}$ denotes the state space and $\langle t \rangle$ is the special start/stop state. Backpropagation uses the identity $\nabla_{\log p}^{\mathcal{L}} = p \odot \nabla_p^{\mathcal{L}}$ to calculate $\nabla_\theta^{\mathcal{L}} = \nabla_{\log p}^{\mathcal{L}} \nabla_\theta^{\log p}$, where $\odot$ is the element-wise multiplication. Typically the forward-backward with marginals is performed in the log-space semifield $\mathbb{R} \cup \{\pm\infty\}$ with binary operations $\oplus = \text{logadd}$ and $\otimes = +$ for numerical precision. However, backpropagation requires working with the log of negative values (since $\nabla_p^{\mathcal{L}}$ could be negative), so we extend to a field $[\mathbb{R} \cup \{\pm\infty\}] \times \{+, -\}$ with special $+/-$ log-space operations. Binary operations applied to vectors are implied to be element-wise. The signexp function is defined as $\text{signexp}(l_a) = s_a \exp(l_a)$. See Section 3.3 and Table 1 for more details.

$x = [x_1, \ldots, x_n]$ is,

$$p(z \,|\, x, q) = \text{softmax} \left( \mathbb{1}\{z \text{ is valid}\} \sum_{i \neq j} \mathbb{1}\{z_{ij} = 1\}\theta_{ij} \right) \tag{4}$$

where $z$ is represented as a vector of $z_{ij}$'s for all $i \neq j$. It is possible to calculate the marginal probability of each edge $p(z_{ij} = 1 \,|\, x, q)$ for all $i, j$ in $O(n^3)$ time using the inside-outside algorithm (Baker, 1979) on the data structures of Eisner (1996).

The parsing contraints ensure that each word has exactly one head (i.e. $\sum_{i=1}^n z_{ij} = 1$). Therefore if we want to utilize the *soft-head* selection of a position $j$, the context vector is defined as:

$$f_j(x, z) = \sum_{i=1}^n \mathbb{1}\{z_{ij} = 1\}\mathbf{x}_i \qquad \mathbf{c}_j = \mathbb{E}_z[f_j(x, z)] = \sum_{i=1}^n p(z_{ij} = 1 \,|\, x, q)\mathbf{x}_i$$

Note that in this case the annotation function has the subscript $j$ to produce a context vector for each word in the sentence. Similar types of attention can be applied for other tree properties (e.g. soft-children). We refer to this type of attention layer as a *syntactic attention* layer.

## 3.3 END-TO-END TRAINING

Graphical models of this form have been widely used as the final layer of deep models. Our contribution is to argue that these networks can be added within deep networks in place of simple attention layers. The whole model can then be trained end-to-end.

The main complication in utilizing this approach within the network itself is the need to backpropagate the gradients through an inference algorithm as part of the structured attention network. Past work has demonstrated the techniques necessary for this approach (see Stoyanov et al. (2011)), but to our knowledge it is very rarely employed.

Consider the case of the simple linear-chain CRF layer from equation (3). Figure 2 (left) shows the standard forward-backward algorithm for computing the marginals $p(z_i = 1 \,|\, x, q; \theta)$. If we treat the forward-backward algorithm as a neural network layer, its input are the potentials $\theta$, and its output

after the forward pass are these marginals.[3] To backpropagate a loss through this layer we need to compute the gradient of the loss $\mathcal{L}$ with respect to $\theta$, $\nabla_\theta^\mathcal{L}$, as a function of the gradient of the loss with respect to the marginals, $\nabla_p^\mathcal{L}$.[4] As the forward-backward algorithm consists of differentiable steps, this function can be derived using reverse-mode automatic differentiation of the forward-backward algorithm itself. Note that this reverse-mode algorithm conveniently has a parallel structure to the forward version, and can also be implemented using dynamic programming.

However, in practice, one cannot simply use current off-the-shelf tools for this task. For one, efficiency is quite important for these models and so the benefits of hand-optimizing the reverse-mode implementation still outweighs simplicity of automatic differentiation. Secondly, numerical precision becomes a major issue for structured attention networks. For computing the forward-pass and the marginals, it is important to use the standard log-space semi-field over $\mathbb{R} \cup \{\pm\infty\}$ with binary operations ($\oplus = \mathrm{logadd}, \otimes = +$) to avoid underflow of probabilities. For computing the backward-pass, we need to remain in log-space, but also handle log of negative values (since $\nabla_p^\mathcal{L}$ could be negative). This requires extending to the *signed* log-space semifield over $[\mathbb{R} \cup \{\pm\infty\}] \times \{+, -\}$ with special $+/-$ operations. Table 1, based on Li & Eisner (2009), demonstrates how to handle this issue, and Figure 2 (right) describes backpropagation through the forward-backward algorithm. For dependency parsing, the forward pass can be computed using the inside-outside implementation of Eisner's algorithm (Eisner, 1996). Similarly, the backpropagation parallels the inside-outside structure. Forward/backward pass through the inside-outside algorithm is described in Appendix B.

| $s_a$ | $s_b$ | $\oplus$ | | $\otimes$ | |
|---|---|---|---|---|---|
| | | $l_{a+b}$ | $s_{a+b}$ | $l_{a \cdot b}$ | $s_{a \cdot b}$ |
| $+$ | $+$ | $l_a + \log(1 + d)$ | $+$ | $l_a + l_b$ | $+$ |
| $+$ | $-$ | $l_a + \log(1 - d)$ | $+$ | $l_a + l_b$ | $-$ |
| $-$ | $+$ | $l_a + \log(1 - d)$ | $-$ | $l_a + l_b$ | $-$ |
| $-$ | $-$ | $l_a + \log(1 + d)$ | $-$ | $l_a + l_b$ | $+$ |

**Table 1:** Signed log-space semifield (from Li & Eisner (2009)). Each real number $a$ is represented as a pair $(l_a, s_a)$ where $l_a = \log|a|$ and $s_a = \mathrm{sign}(a)$. Therefore $a = s_a \exp(l_a)$. For the above we let $d = \exp(l_b - l_a)$ and assume $|a| > |b|$.

## 4 EXPERIMENTS

We experiment with three instantiations of structured attention networks on four different tasks: (a) a simple, synthetic tree manipulation task using the syntactic attention layer, (b) machine translation with segmentation attention (i.e. two-state linear-chain CRF), (c) question answering using an $n$-state linear-chain CRF for multi-step inference over $n$ facts, and (d) natural language inference with syntactic tree attention. These experiments are not intended to boost the state-of-the-art for these tasks but to test whether these methods can be trained effectively in an end-to-end fashion, can yield improvements over standard selection-based attention, and can learn plausible latent structures. All model architectures, hyperparameters, and training details are further described in Appendix A.

### 4.1 TREE TRANSDUCTION

The first set of experiments look at a tree-transduction task. These experiments use synthetic data to explore a failure case of soft-selection attention models. The task is to learn to convert a random formula given in prefix notation to one in infix notation, e.g.,

$( * ( + ( + 15 \ 7 ) 1 \ 8 ) ( + 19 \ 0 \ 11 ) ) \Rightarrow ( ( 15 + 7 ) + 1 + 8 ) * ( 19 + 0 + 11 )$

The alphabet consists of symbols $\{(, ), +, *\}$, numbers between 0 and 20, and a special root symbol \$. This task is used as a preliminary task to see if the model is able to learn the implicit tree structure on the source side. The model itself is an encoder-decoder model, where the encoder is defined below and the decoder is an LSTM. See Appendix A.2 for the full model.

---

[3]Confusingly, "forward" in this case is different than in the *forward*-backward algorithm, as the marginals themselves are the output. However the two uses of the term are actually quite related. The forward-backward algorithm can be interpreted as a forward and backpropagation pass on the log partition function. See Eisner (2016) for further details (appropriately titled "Inside-Outside and Forward-Backward Algorithms Are Just Backprop"). As such our full approach can be seen as computing second-order information. This interpretation is central to Li & Eisner (2009).

[4]In general we use $\nabla_b^a$ to denote the Jacobian of $a$ with respect to $b$.

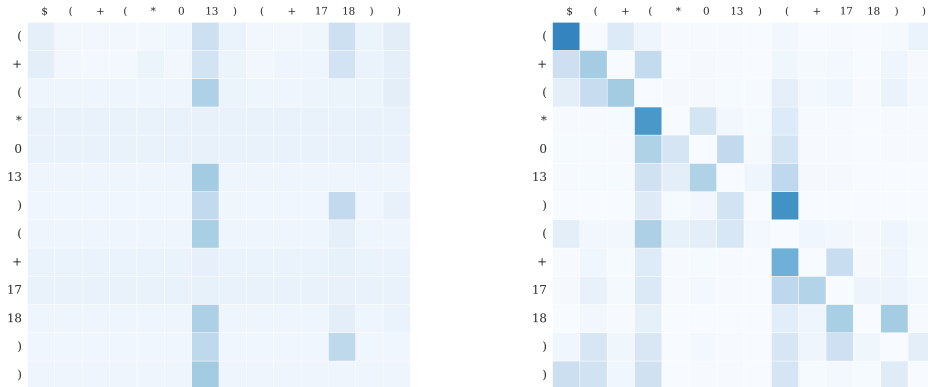

**Figure 3:** Visualization of the source self-attention distribution for the simple (left) and structured (right) attention models on the tree transduction task. $ is the special root symbol. Each row delineates the distribution over the parents (i.e. each row sums to one). The attention distribution obtained from the parsing marginals are more able to capture the tree structure—e.g. the attention weights of closing parentheses are generally placed on the opening parentheses (though not necessarily on a single parenthesis).

Training uses 15K prefix-infix pairs where the maximum nesting depth is set to be between 2-4 (the above example has depth 3), with 5K pairs in each depth bucket. The number of expressions in each parenthesis is limited to be at most 4. Test uses 1K unseen sequences with depth between 2-6 (note specifically deeper than train), with 200 sequences for each depth. The performance is measured as the average proportion of correct target tokens produced until the first failure (as in Grefenstette et al. (2015)).

For experiments we try using different forms of *self*-attention over embedding-only encoders. Let $\mathbf{x}_j$ be an embedding for each source symbol; our three variants of the source representation $\hat{\mathbf{x}}_j$ are: (a) *no atten*, just symbol embeddings by themselves, i.e. $\hat{\mathbf{x}}_j = \mathbf{x}_j$; (b) *simple* attention, symbol embeddings and soft-pairing for each symbol, i.e. $\hat{\mathbf{x}}_j = [\mathbf{x}_j; \mathbf{c}_j]$ where $\mathbf{c}_j = \sum_{i=1}^{n} \text{softmax}(\theta_{ij})\mathbf{x}_i$ is calculated using soft-selection; (c) *structured* attention, symbol embeddings and soft-parent, i.e. $\hat{\mathbf{x}}_j = [\mathbf{x}_j; \mathbf{c}_j]$ where $\mathbf{c}_j = \sum_{i=1}^{n} p(z_{ij} = 1 \,|\, x)\mathbf{x}_i$ is calculated using parsing marginals, obtained from the syntactic attention layer. None of these models use an explicit query value—the potentials come from running a bidirectional LSTM over the source, producing hidden vectors $\mathbf{h}_i$, and then computing

$$\theta_{ij} = \tanh(\mathbf{s}^{\top} \tanh(\mathbf{W}_1 \mathbf{h}_i + \mathbf{W}_2 \mathbf{h}_j + \mathbf{b}))$$

where $\mathbf{s}, \mathbf{b}, \mathbf{W}_1, \mathbf{W}_2$ are parameters (see Appendix A.1).

| Depth | No Atten | Simple | Structured |
|-------|----------|--------|------------|
| 2 | 7.6 | 87.4 | 99.2 |
| 3 | 4.1 | 49.6 | 87.0 |
| 4 | 2.8 | 23.3 | 64.5 |
| 5 | 2.1 | 15.0 | 30.8 |
| 6 | 1.5 | 8.5 | 18.2 |

**Table 2:** Performance (average length to failure %) of models on the tree-transduction task.

The source representation $[\hat{\mathbf{x}}_1, \ldots, \hat{\mathbf{x}}_n]$ are attended over using the standard attention mechanism at each decoding step by an LSTM decoder.[5] Additionally, symbol embedding parameters are shared between the parsing LSTM and the source encoder.

**Results** Table 2 has the results for the task. Note that this task is fairly difficult as the encoder is quite simple. The baseline model (unsurprisingly) performs poorly as it has no information about the source ordering. The simple attention model performs better, but is significantly outperformed by the structured model with a tree structure bias. We hypothesize that the model is partially reconstructing the arithmetic tree. Figure 3 shows the attention distribution for the simple/structured models on the same source sequence, which indicates that the structured model is able to learn boundaries (i.e. parentheses).

---

[5]Thus there are two attention mechanisms at work under this setup. First, structured attention over the source only to obtain soft-parents for each symbol (i.e. self-attention). Second, standard softmax alignment attention over the source representations during decoding.

### 4.2 NEURAL MACHINE TRANSLATION

Our second set of experiments use a full neural machine translation model utilizing attention over subsequences. Here both the encoder/decoder are LSTMs, and we replace standard simple attention with a segmentation attention layer. We experiment with two settings: translating directly from unsegmented Japanese characters to English words (effectively using structured attention to perform soft word segmentation), and translating from segmented Japanese words to English words (which can be interpreted as doing *phrase-based* neural machine translation). Japanese word segmentation is done using the KyTea toolkit (Neubig et al., 2011).

The data comes from the Workshop on Asian Translation (WAT) (Nakazawa et al., 2016). We randomly pick 500K sentences from the original training set (of 3M sentences) where the Japanese sentence was at most 50 characters and the English sentence was at most 50 words. We apply the same length filter on the provided validation/test sets for evaluation. The vocabulary consists of all tokens that occurred at least 10 times in the training corpus.

The segmentation attention layer is a two-state CRF where the unary potentials at the $j$-th decoder step are parameterized as

$$\theta_i(k) = \begin{cases} \mathbf{h}_i \mathbf{W} \mathbf{h}_j, & k = 1 \\ 0, & k = 0 \end{cases}$$

Here $[\mathbf{h}_1, \ldots, \mathbf{h}_n]$ are the encoder hidden states and $\mathbf{h}'_j$ is the $j$-th decoder hidden state (i.e. the query vector). The pairwise potentials are parameterized linearly with $\mathbf{b}$, i.e. all together

$$\theta_{i,i+1}(z_i, z_{i+1}) = \theta_i(z_i) + \theta_{i+1}(z_{i+1}) + \mathbf{b}_{z_i, z_{i+1}}$$

Therefore the segmentation attention layer requires just $4$ additional parameters. Appendix A.3 describes the full model architecture.

We experiment with three attention configurations: (a) standard *simple* attention, i.e. $\mathbf{c}_j = \sum_{i=1}^n \text{softmax}(\theta_i)\mathbf{h}_i$; (b) *sigmoid* attention: multiple selection with Bernoulli random variables, i.e. $\mathbf{c}_j = \sum_{i=1}^n \text{sigmoid}(\theta_i)\mathbf{h}_i$; (c) *structured* attention, encoded with normalized CRF marginals,

$$\mathbf{c}_j = \sum_{i=1}^n \frac{p(z_i = 1 \mid x, q)}{\gamma} \mathbf{h}_i \qquad \qquad \gamma = \frac{1}{\lambda} \sum_{i=1}^n p(z_i = 1 \mid x, q)$$

The normalization term $\gamma$ is not ideal but we found it to be helpful for stable training.[6] $\lambda$ is a hyperparameter (we use $\lambda = 2$) and we further add an $l_2$ penalty of $0.005$ on the pairwise potentials $\mathbf{b}$. These values were found via grid search on the validation set.

|       | Simple | Sigmoid | Structured |
|-------|--------|---------|------------|
| CHAR  | 12.6   | 13.1    | 14.6       |
| WORD  | 14.1   | 13.8    | 14.3       |

**Table 3:** Translation performance as measured by BLEU (higher is better) on character-to-word and word-to-word Japanese-English translation for the three different models.

**Results** Results for the translation task on the test set are given in Table 3. Sigmoid attention outperforms simple (softmax) attention on the character-to-word task, potentially because it is able to learn many-to-one alignments. On the word-to-word task, the opposite is true, with simple attention outperforming sigmoid attention. Structured attention outperforms both models on both tasks, although improvements on the word-to-word task are modest and unlikely to be statistically significant.

For further analysis, Figure 4 shows a visualization of the different attention mechanisms on the character-to-word setup. The simple model generally focuses attention heavily on a single character. In contrast, the sigmoid and structured models are able to spread their attention distribution on contiguous subsequences. The structured attention learns additional parameters (i.e. $\mathbf{b}$) to smooth out this type of attention.

---

[6]With standard expectation (i.e. $\mathbf{c}_j = \sum_{i=1}^n p(z_i = 1 \mid x, q)\mathbf{h}_i$) we empirically observed the marginals to quickly saturate. We tried various strategies to overcome this, such as putting an $l_2$ penalty on the unary potentials and initializing with a pretrained sigmoid attention model, but simply normalizing the marginals proved to be the most effective. However, this changes the interpretation of the context vector as the expectation of an annotation function in this case.

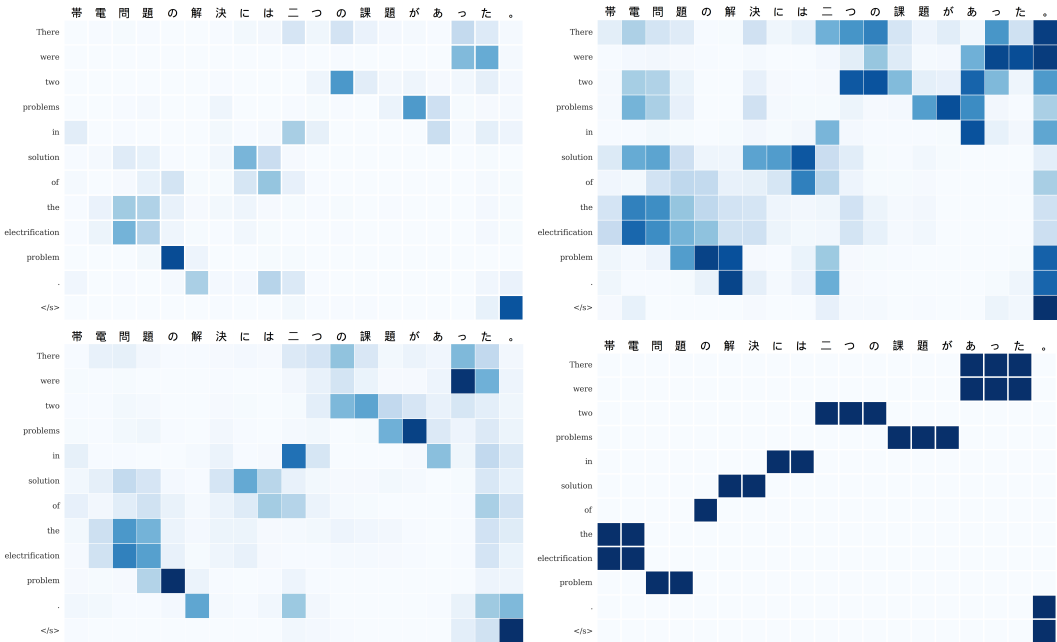

**Figure 4:** Visualization of the source attention distribution for the simple (top left), sigmoid (top right), and structured (bottom left) attention models over the ground truth sentence on the character-to-word translation task. Manually-annotated alignments are shown in bottom right. Each row delineates the attention weights over the source sentence at each step of decoding. The sigmoid/structured attention models are able learn an implicit segmentation model and focus on multiple characters at each time step.

## 4.3 QUESTION ANSWERING

Our third experiment is on question answering (QA) with the linear-chain CRF attention layer for inference over multiple facts. We use the bAbI dataset (Weston et al., 2015), where the input is a set of sentences/facts paired with a question, and the answer is a single token. For many of the tasks the model has to attend to multiple supporting facts to arrive at the correct answer (see Figure 5 for an example), and existing approaches use multiple 'hops' to greedily attend to different facts. We experiment with employing structured attention to perform inference in a non-greedy way. As the ground truth supporting facts are given in the dataset, we are able to assess the model's inference accuracy.

The baseline (simple) attention model is the End-To-End Memory Network (Sukhbaatar et al., 2015) (MemN2N), which we briefly describe here. See Appendix A.4 for full model details. Let $\mathbf{x}_1, \ldots, \mathbf{x}_n$ be the input embedding vectors for the $n$ sentences/facts and let $\mathbf{q}$ be the query embedding. In MemN2N, $z_k$ is the random variable for the sentence to select at the $k$-th inference step (i.e. $k$-th hop), and thus $z_k \in \{1, \ldots, n\}$. The probability distribution over $z_k$ is given by $p(z_k = i \mid x, q) = \text{softmax}((\mathbf{x}_i^k)^\top \mathbf{q}^k)$, and the context vector is given by $\mathbf{c}^k = \sum_{i=1}^n p(z_k = i \mid x, q) \mathbf{o}_i^k$, where $\mathbf{x}_i^k, \mathbf{o}_i^k$ are the input and output embedding for the $i$-th sentence at the $k$-th hop, respectively. The $k$-th context vector is used to modify the query $\mathbf{q}^{k+1} = \mathbf{q}^k + \mathbf{c}^k$, and this process repeats for $k = 1, \ldots, K$ (for $k = 1$ we have $\mathbf{x}_i^k = \mathbf{x}_i, \mathbf{q}^k = \mathbf{q}, \mathbf{c}^k = \mathbf{0}$). The $K$-th context and query vectors are used to obtain the final answer. The attention mechanism for a $K$-hop MemN2N network can therefore be interpreted as a greedy selection of a length-$K$ sequence of facts (i.e. $z_1, \ldots, z_K$).

For structured attention, we use an $n$-state, $K$-step linear-chain CRF.[7] We experiment with two different settings: (a) a unary CRF model with node potentials

$$\theta_k(i) = (\mathbf{x}_i^k)^\top \mathbf{q}^k$$

---

[7]Note that this differs from the segmentation attention for the neural machine translation experiments described above, which was a $K$-state (with $K = 2$), $n$-step linear-chain CRF.

| Task | $K$ | MemN2N Ans % | MemN2N Fact % | Binary CRF Ans % | Binary CRF Fact % | Unary CRF Ans % | Unary CRF Fact % |
|---|---|---|---|---|---|---|---|
| TASK 02 - TWO SUPPORTING FACTS | 2 | 87.3 | 46.8 | 84.7 | 81.8 | 43.5 | 22.3 |
| TASK 03 - THREE SUPPORTING FACTS | 3 | 52.6 | 1.4 | 40.5 | 0.1 | 28.2 | 0.0 |
| TASK 07 - COUNTING | 3 | 83.2 | — | 83.5 | — | 79.3 | — |
| TASK 08 - LISTS SETS | 3 | 94.1 | — | 93.3 | — | 87.1 | — |
| TASK 11 - INDEFINITE KNOWLEDGE | 2 | 97.8 | 38.2 | 97.7 | 80.8 | 88.6 | 0.0 |
| TASK 13 - COMPOUND COREFERENCE | 2 | 95.6 | 14.8 | 97.0 | 36.4 | 94.4 | 9.3 |
| TASK 14 - TIME REASONING | 2 | 99.9 | 77.6 | 99.7 | 98.2 | 90.5 | 30.2 |
| TASK 15 - BASIC DEDUCTION | 2 | 100.0 | 59.3 | 100.0 | 89.5 | 100.0 | 51.4 |
| TASK 16 - BASIC INDUCTION | 3 | 97.1 | 91.0 | 97.9 | 85.6 | 98.0 | 41.4 |
| TASK 17 - POSITIONAL REASONING | 2 | 61.1 | 23.9 | 60.6 | 49.6 | 59.7 | 10.5 |
| TASK 18 - SIZE REASONING | 2 | 86.4 | 3.3 | 92.2 | 3.9 | 92.0 | 1.4 |
| TASK 19 - PATH FINDING | 2 | 21.3 | 10.2 | 24.4 | 11.5 | 24.3 | 7.8 |
| AVERAGE | — | 81.4 | 39.6 | 81.0 | 53.7 | 73.8 | 17.4 |

**Table 4:** Answer accuracy (Ans %) and supporting fact selection accuracy (Fact %) of the three QA models on the 1K bAbI dataset. $K$ indicates the number of hops/inference steps used for each task. Task 7 and 8 both contain variable number of facts and hence they are excluded from the fact accuracy measurement. Supporting fact selection accuracy is calculated by taking the average of 10 best runs (out of 20) for each task.

and (b) a binary CRF model with pairwise potentials

$$\theta_{k,k+1}(i,j) = (\mathbf{x}_i^k)^\top \mathbf{q}^k + (\mathbf{x}_i^k)^\top \mathbf{x}_j^{k+1} + (\mathbf{x}_j^{k+1})^\top \mathbf{q}^{k+1}$$

The binary CRF model is designed to test the model's ability to perform sequential reasoning. For both (a) and (b), a *single* context vector is computed: $\mathbf{c} = \sum_{z_1,\ldots,z_K} p(z_1,\ldots,z_K \,|\, x, q) f(x, z)$ (unlike MemN2N which computes $K$ context vectors). Evaluating $\mathbf{c}$ requires summing over all $n^K$ possible sequences of length $K$, which may not be practical for large values of $K$. However, if $f(x, z)$ factors over the components of $z$ (e.g. $f(x, z) = \sum_{k=1}^{K} f_k(x, z_k)$) then one can rewrite the above sum in terms of marginals: $\mathbf{c} = \sum_{k=1}^{K} \sum_{i=1}^{n} p(z_k = i \,|\, x, q) f_k(x, z_k)$. In our experiments, we use $f_k(x, z_k) = \mathbf{o}_{z_k}^k$. All three models are described in further detail in Appendix A.4.

**Results** We use the version of the dataset with 1K questions for each task. Since all models reduce to the same network for tasks with 1 supporting fact, they are excluded from our experiments. The number of hops (i.e. $K$) is task-dependent, and the number of memories (i.e. $n$) is limited to be at most 25 (note that many question have less than 25 facts—e.g. the example in Figure 5 has 9 facts). Due to high variance in model performance, we train 20 models with different initializations for each task and report the test accuracy of the model that performed the best on a 10% held-out validation set (as is typically done for bAbI tasks).

Results of the three different models are shown in Table 4. For correct answer seletion (Ans %), we find that MemN2N and the Binary CRF model perform similarly while the Unary CRF model does worse, indicating the importance of including pairwise potentials. We also assess each model's ability to attend to the correct supporting facts in Table 4 (Fact %). Since ground truth supporting facts are provided for each query, we can check the sequence accuracy of supporting facts for each model (i.e. the rate of selecting the exact correct sequence of facts) by taking the highest probability sequence $\hat{z} = \arg\max p(z_1,\ldots,z_K \,|\, x, q)$ from the model and checking against the ground truth. Overall the Binary CRF is able to recover supporting facts better than MemN2N. This improvement is significant and can be up to two-fold as seen for task 2, 11, 13 & 17. However we observed that on many tasks it is sufficient to select only the last (or first) fact correctly to predict the answer, and thus higher sequence selection accuracy does not necessarily imply better answer accuracy (and vice versa). For example, all three models get 100% answer accuracy on task 15 but have different supporting fact accuracies.

Finally, in Figure 5 we visualize of the output edge marginals produced by the Binary CRF model for a single question in task 16. In this instance, the model is uncertain but ultimately able to select the right sequence of facts $5 \rightarrow 6 \rightarrow 8$.

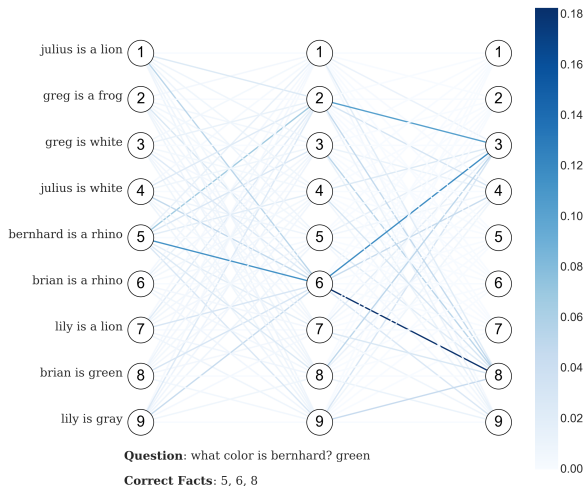

**Figure 5:** Visualization of the attention distribution over supporting fact sequences for an example question in task 16 for the Binary CRF model. The actual question is displayed at the bottom along with the correct answer and the ground truth supporting facts ($5 \rightarrow 6 \rightarrow 8$). The edges represent the marginal probabilities $p(z_k, z_{k+1} \mid x, q)$, and the nodes represent the $n$ supporting facts (here we have $n = 9$). The text for the supporting facts are shown on the left. The top three most likely sequences are: $p(z_1 = 5, z_2 = 6, z_3 = 8 \mid x, q) = 0.0564$, $p(z_1 = 5, z_2 = 6, z_3 = 3 \mid x, q) = 0.0364$, $p(z_1 = 5, z_2 = 2, z_3 = 3 \mid x, q) = 0.0356$.

## 4.4 NATURAL LANGUAGE INFERENCE

The final experiment looks at the task of natural language inference (NLI) with the syntactic attention layer. In NLI, the model is given two sentences (hypothesis/premise) and has to predict their relationship: entailment, contradiction, neutral.

For this task, we use the Stanford NLI dataset (Bowman et al., 2015) and model our approach off of the decomposable attention model of Parikh et al. (2016). This model takes in the matrix of word embeddings as the input for each sentence and performs *inter-sentence* attention to predict the answer. Appendix A.5 describes the full model.

As in the transduction task, we focus on modifying the input representation to take into account soft parents via self-attention (i.e. *intra-sentence* attention). In addition to the three baselines described for tree transduction (No Attention, Simple, Structured), we also explore two additional settings: (d) *hard* pipeline parent selection, i.e. $\hat{\mathbf{x}}_j = [\mathbf{x}_j; \mathbf{x}_{\text{head}(j)}]$, where $\text{head}(j)$ is the index of $x_j$'s parent[8]; (e) *pretrained* structured attention: structured attention where the parsing layer is pretrained for one epoch on a parsed dataset (which was enough for convergence).

**Results** Results of our models are shown in Table 5. Simple attention improves upon the no attention model, and this is consistent with improvements observed by Parikh et al. (2016) with their intra-sentence attention model. The pipelined model with hard parents also slightly improves upon the baseline. Structured attention outperforms both models, though surprisingly, pretraining the syntactic attention layer on the parse trees performs worse than training it from scratch—it is possible that the pretrained attention is too strict for this task.

We also obtain the hard parse for an example sentence by running the Viterbi algorithm on the syntactic attention layer with the non-pretrained model:

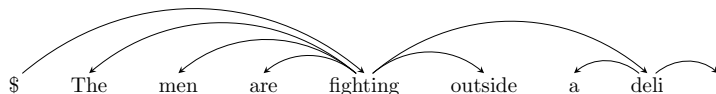

---

[8]The parents are obtained from running the dependency parser of Andor et al. (2016), available at
`https://github.com/tensorflow/models/tree/master/syntaxnet`

| Model | Accuracy % |
|---|---|
| Handcrafted features (Bowman et al., 2015) | 78.2 |
| LSTM encoders (Bowman et al., 2015) | 80.6 |
| Tree-Based CNN (Mou et al., 2016) | 82.1 |
| Stack-Augmented Parser-Interpreter Neural Net (Bowman et al., 2016) | 83.2 |
| LSTM with word-by-word attention (Rocktäschel et al., 2016) | 83.5 |
| Matching LSTMs (Wang & Jiang, 2016) | 86.1 |
| Decomposable attention over word embeddings (Parikh et al., 2016) | 86.3 |
| Decomposable attention + intra-sentence attention (Parikh et al., 2016) | 86.8 |
| Attention over constituency tree nodes (Zhao et al., 2016) | 87.2 |
| Neural Tree Indexers (Munkhdalai & Yu, 2016) | 87.3 |
| Enhanced BiLSTM Inference Model (Chen et al., 2016) | 87.7 |
| Enhanced BiLSTM Inference Model + ensemble (Chen et al., 2016) | 88.3 |
| No Attention | 85.8 |
| No Attention + Hard parent | 86.1 |
| Simple Attention | 86.2 |
| Structured Attention | 86.8 |
| Pretrained Structured Attention | 86.5 |

**Table 5:** Results of our models (bottom) and others (top) on the Stanford NLI test set. Our baseline model has the same architecture as Parikh et al. (2016) but the performance is slightly different due to different settings (e.g. we train for 100 epochs with a batch size of 32 while Parikh et al. (2016) train for 400 epochs with a batch size of 4 using asynchronous SGD.)

Despite being trained without ever being exposed to an explicit parse tree, the syntactic attention layer learns an almost plausible dependency structure. In the above example it is able to correctly identify the main verb `fighting`, but makes mistakes on determiners (e.g. head of `The` should be `men`). We generally observed this pattern across sentences, possibly because the verb structure is more important for the inference task.

## 5 CONCLUSION

This work outlines structured attention networks, which incorporate graphical models to generalize simple attention, and describes the technical machinery and computational techniques for backpropagating through models of this form. We implement two classes of structured attention layers: a linear-chain CRF (for neural machine translation and question answering) and a more complicated first-order dependency parser (for tree transduction and natural language inference). Experiments show that this method can learn interesting structural properties and improve on top of standard models. Structured attention could also be a way of learning latent labelers or parsers through attention on other tasks.

It should be noted that the additional complexity in computing the attention distribution increases run-time—for example, structured attention was approximately $5\times$ slower to train than simple attention for the neural machine translation experiments, even though both attention layers have the same asymptotic run-time (i.e. $O(n)$).

Embedding *differentiable inference* (and more generally, *differentiable algorithms*) into deep models is an exciting area of research. While we have focused on models that admit (tractable) exact inference, similar technique can be used to embed approximate inference methods. Many optimization algorithms (e.g. gradient descent, LBFGS) are also differentiable (Domke, 2012; Maclaurin et al., 2015), and have been used as output layers for structured prediction in energy-based models (Belanger & McCallum, 2016; Wang et al., 2016). Incorporating them as internal neural network layers is an interesting avenue for future work.

ACKNOWLEDGMENTS

We thank Tao Lei, Ankur Parikh, Tim Vieira, Matt Gormley, André Martins, Jason Eisner, Yoav Goldberg, and the anonymous reviewers for helpful comments, discussion, notes, and code. We additionally thank Yasumasa Miyamoto for verifying Japanese-English translations.

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

APPENDICES

# A MODEL DETAILS

## A.1 SYNTACTIC ATTENTION

The syntactic attention layer (for tree transduction and natural language inference) is similar to the first-order graph-based dependency parser of Kipperwasser & Goldberg (2016). Given an input sentence $[x_1, \ldots, x_n]$ and the corresponding word vectors $[\mathbf{x}_1, \ldots, \mathbf{x}_n]$, we use a bidirectional LSTM to get the hidden states for each time step $i \in [1, \ldots, n]$,

$$\mathbf{h}_i^{\text{fwd}} = \text{LSTM}(\mathbf{x}_i, \mathbf{h}_{i-1}^{\text{fwd}}) \qquad \mathbf{h}_i^{\text{bwd}} = \text{LSTM}(\mathbf{x}_i, \mathbf{h}_{i+1}^{\text{bwd}}) \qquad \mathbf{h}_i = [\mathbf{h}_i^{\text{fwd}}; \mathbf{h}_i^{\text{bwd}}]$$

where the forward and backward LSTMs have their own parameters. The score for $x_i \to x_j$ (i.e. $x_i$ is the parent of $x_j$), is given by an MLP

$$\theta_{ij} = \tanh(\mathbf{s}^\top \tanh(\mathbf{W}_1 \mathbf{h}_i + \mathbf{W}_2 \mathbf{h}_j + \mathbf{b}))$$

These scores are used as input to the inside-outside algorithm (see Appendix B) to obtain the probability of each word's parent $p(z_{ij} = 1 \,|\, x)$, which is used to obtain the soft-parent $\mathbf{c}_j$ for each word $x_j$. In the non-structured case we simply have $p(z_{ij} = 1 \,|\, x) = \text{softmax}(\theta_{ij})$.

## A.2 TREE TRANSDUCTION

Let $[x_1, \ldots, x_n], [y_1, \ldots, y_m]$ be the sequence of source/target symbols, with the associated embeddings $[\mathbf{x}_1, \ldots, \mathbf{x}_n], [\mathbf{y}_1, \ldots, \mathbf{y}_m]$ with $\mathbf{x}_i, \mathbf{y}_j \in \mathbb{R}^l$. In the simplest baseline model we take the source representation to be the matrix of the symbol embeddings. The decoder is a one-layer LSTM which produces the hidden states $\mathbf{h}'_j = \text{LSTM}(\mathbf{y}_j, \mathbf{h}'_{j-1})$, with $\mathbf{h}'_j \in \mathbb{R}^l$. The hidden states are combined with the input representation via a bilinear map $\mathbf{W} \in \mathbb{R}^{l \times l}$ to produce the attention distribution used to obtain the vector $\mathbf{m}_i$, which is combined with the decoder hidden state as follows,

$$\alpha_i = \frac{\exp \mathbf{x}_i \mathbf{W} \mathbf{h}'_j}{\sum_{k=1}^n \exp \mathbf{x}_k \mathbf{W} \mathbf{h}'_j} \qquad \mathbf{m}_i = \sum_{i=1}^n \alpha_i \mathbf{x}_i \qquad \hat{\mathbf{h}}_j = \tanh(\mathbf{U}[\mathbf{m}_i; \mathbf{h}'_j])$$

Here we have $\mathbf{W} \in \mathbb{R}^{l \times l}$ and $\mathbf{U} \in \mathbb{R}^{2l \times l}$. Finally, $\hat{\mathbf{h}}_j$ is used to to obtain a distribution over the next symbol $y_{j+1}$,

$$p(y_{j+1} \,|\, x_1, \ldots, x_n, y_1, \ldots, y_j) = \text{softmax}(\mathbf{V}\hat{\mathbf{h}}_j + \mathbf{b})$$

For structured/simple models, the $j$-th source representation are respectively

$$\hat{\mathbf{x}}_i = \left[ \mathbf{x}_i; \sum_{k=1}^n p(z_{ki} = 1 \,|\, x) \mathbf{x}_k \right] \qquad \hat{\mathbf{x}}_i = \left[ \mathbf{x}_i; \sum_{k=1}^n \text{softmax}(\theta_{ki}) \mathbf{x}_k \right]$$

where $\theta_{ij}$ comes from the bidirectional LSTM described in A.1. Then $\alpha_i$ and $\mathbf{m}_i$ changed accordingly,

$$\alpha_i = \frac{\exp \hat{\mathbf{x}}_i \mathbf{W} \mathbf{h}'_j}{\sum_{k=1}^n \exp \hat{\mathbf{x}}_k \mathbf{W} \mathbf{h}'_j} \qquad \mathbf{m}_i = \sum_{i=1}^n \alpha_i \hat{\mathbf{x}}_i$$

Note that in this case we have $\mathbf{W} \in \mathbb{R}^{2l \times l}$ and $\mathbf{U} \in \mathbb{R}^{3l \times l}$. We use $l = 50$ in all our experiments. The forward/backward LSTMs for the parsing LSTM are also 50-dimensional. Symbol embeddings are shared between the encoder and the parsing LSTMs.

Additional training details include: batch size of 20; training for 13 epochs with a learning rate of 1.0, which starts decaying by half after epoch 9 (or the epoch at which performance does not improve on validation, whichever comes first); parameter initialization over a uniform distribution $U[-0.1, 0.1]$; gradient normalization at 1 (i.e. renormalize the gradients to have norm 1 if the $l_2$ norm exceeds 1). Decoding is done with beam search (beam size $= 5$).

A.3    NEURAL MACHINE TRANSLATION

The baseline NMT system is from Luong et al. (2015). Let $[x_1, \ldots, x_n], [y_1, \ldots, y_m]$ be the source/target sentence, with the associated word embeddings $[\mathbf{x}_1, \ldots, \mathbf{x}_n], [\mathbf{y}_1, \ldots, \mathbf{y}_m]$. The encoder is an LSTM over the source sentence, which produces the hidden states $[\mathbf{h}_1, \ldots, \mathbf{h}_n]$ where

$$\mathbf{h}_i = \text{LSTM}(\mathbf{x}_i, \mathbf{h}_{i-1})$$

and $\mathbf{h}_i \in \mathbb{R}^l$. The decoder is another LSTM which produces the hidden states $\mathbf{h}'_j \in \mathbb{R}^l$. In the simple attention case with categorical attention, the hidden states are combined with the input representation via a bilinear map $\mathbf{W} \in \mathbb{R}^{l \times l}$ and this distribution is used to obtain the context vector at the $j$-th time step,

$$\theta_i = \mathbf{h}_i \mathbf{W} \mathbf{h}'_j \qquad\qquad \mathbf{c}_j = \sum_{i=1}^{n} \text{softmax}(\theta_i) \mathbf{h}_i$$

The Bernoulli attention network has the same $\theta_i$ but instead uses a sigmoid to obtain the weights of the linear combination, i.e.,

$$\mathbf{c}_j = \sum_{i=1}^{n} \text{sigmoid}(\theta_i) \mathbf{h}_i$$

And finally, the structured attention model uses a bilinear map to parameterize one of the unary potentials

$$\theta_i(k) = \begin{cases} \mathbf{h}_i \mathbf{W} \mathbf{h}'_j, & k = 1 \\ 0, & k = 0 \end{cases}$$

$$\theta_{i,i+1}(z_i, z_{i+1}) = \theta_i(z_i) + \theta_{i+1}(z_{i+1}) + \mathbf{b}_{z_i, z_{i+1}}$$

where $\mathbf{b}$ are the pairwise potentials. These potentials are used as inputs to the forward-backward algorithm to obtain the marginals $p(z_i = 1 \mid x, q)$, which are further normalized to obtain the context vector

$$\mathbf{c}_j = \sum_{i=1}^{n} \frac{p(z_i = 1 \mid x, q)}{\gamma} \mathbf{h}_i \qquad\qquad \gamma = \frac{1}{\lambda} \sum_{i}^{n} p(z_i = 1 \mid x, q)$$

We use $\lambda = 2$ and also add an $l_2$ penalty of $0.005$ on the pairwise potentials $\mathbf{b}$. The context vector is then combined with the decoder hidden state

$$\hat{\mathbf{h}}_j = \tanh(\mathbf{U}[\mathbf{c}_j; \mathbf{h}'_j])$$

and $\hat{\mathbf{h}}_j$ is used to obtain the distribution over the next target word $y_{j+1}$

$$p(y_{j+1} \mid x_1, \ldots, x_n, y_1, \ldots y_j) = \text{softmax}(\mathbf{V}\hat{\mathbf{h}}_j + \mathbf{b})$$

The encoder/decoder LSTMs have 2 layers and 500 hidden units (i.e. $l = 500$).

Additional training details include: batch size of 128; training for 30 epochs with a learning rate of 1.0, which starts decaying by half after the first epoch at which performance does not improve on validation; dropout with probability 0.3; parameter initialization over a uniform distribution $U[-0.1, 0.1]$; gradient normalization at 1. We generate target translations with beam search (beam size = 5), and evaluate with `multi-bleu.perl` from Moses.[9]

A.4    QUESTION ANSWERING

Our baseline model (MemN2N) is implemented following the same architecture as described in Sukhbaatar et al. (2015). In particular, let $x = [x_1, \ldots, x_n]$ represent the sequence of $n$ facts with the associated embeddings $[\mathbf{x}_1, \ldots, \mathbf{x}_n]$ and let $\mathbf{q}$ be the embedding of the query $q$. The embeddings

---

[9]`https://github.com/moses-smt/mosesdecoder/blob/master/scripts/generic/multi-bleu.perl`

are obtained by simply adding the word embeddings in each sentence or query. The full model with $K$ hops is as follows:

$$p(z_k = i \,|\, x, q) = \text{softmax}((\mathbf{x}_i^k)^\top \mathbf{q}^k)$$

$$\mathbf{c}^k = \sum_{i=1}^{n} p(z_k = i \,|\, x, q) \mathbf{o}_i^k$$

$$\mathbf{q}^{k+1} = \mathbf{q}^k + \mathbf{c}^k$$

$$p(y \,|\, x, q) = \text{softmax}(\mathbf{W}(\mathbf{q}^K + \mathbf{c}^K))$$

where $p(y \,|\, x, q)$ is the distribution over the answer vocabulary. At each layer, $\{\mathbf{x}_i^k\}$ and $\{\mathbf{o}_i^k\}$ are computed using embedding matrices $\mathbf{X}^k$ and $\mathbf{O}^k$. We use the *adjacent weight tying scheme* from the paper so that $\mathbf{X}^{k+1} = \mathbf{O}^k, \mathbf{W}^T = \mathbf{O}^K$. $\mathbf{X}^1$ is also used to compute the query embedding at the first hop. For $k = 1$ we have $\mathbf{x}_i^k = \mathbf{x}_i, \mathbf{q}^k = \mathbf{q}, \mathbf{c}^k = \mathbf{0}$.

For both the Unary and the Binary CRF models, the same input fact and query representations are computed (i.e. same embedding matrices with weight tying scheme). For the unary model, the potentials are parameterized as

$$\theta_k(i) = (\mathbf{x}_i^k)^\top \mathbf{q}^k$$

and for the binary model we compute pairwise potentials as

$$\theta_{k,k+1}(i,j) = (\mathbf{x}_i^k)^\top \mathbf{q}^k + (\mathbf{x}_i^k)^\top \mathbf{x}_j^{k+1} + (\mathbf{x}_j^{k+1})^\top \mathbf{q}^{k+1}$$

The $\mathbf{q}^k$'s are updated simply with a linear mapping, i.e.

$$\mathbf{q}^{k+1} = \mathbf{Q}\mathbf{q}^k$$

In the case of the Binary CRF, to discourage the model from selecting the same fact again we additionally set $\theta_{k,k+1}(i,i) = -\infty$ for all $i \in \{1, \ldots, n\}$. Given these potentials, we compute the marginals $p(z_k = i, z_{k+1} = j \,|\, x, q)$ using the forward-backward algorithm, which is then used to compute the context vector:

$$\mathbf{c} = \sum_{z_1, \ldots, z_K} p(z_1, \ldots, z_K \,|\, x, q) f(x, z) \qquad f(x, z) = \sum_{k=1}^{K} f_k(x, z_k) \qquad f_k(x, z_k) = \mathbf{o}_{z_k}^k$$

Note that if $f(x, z)$ factors over the components of $z$ (as is the case above) then computing $\mathbf{c}$ only requires evaluating the marginals $p(z_k \,|\, x, q)$.

Finally, given the context vector the prediction is made in a similar fashion to MemN2N:

$$p(y \,|\, x, q) = \text{softmax}(\mathbf{W}(\mathbf{q}^K + \mathbf{c}))$$

Other training setup is similar to Sukhbaatar et al. (2015): we use stochastic gradient descent with learning rate 0.01, which is divided by 2 every 25 epochs until 100 epochs are reached. Capacity of the memory is limited to 25 sentences. The embedding vectors are of size 20 and gradients are renormalized if the norm exceeds 40. All models implement *position encoding*, *temporal encoding*, and *linear start* from the original paper. For linear start, the $\text{softmax}(\cdot)$ function in the attention layer is removed at the beginning and re-inserted after 20 epochs for MemN2N, while for the CRF models we apply a $\log(\text{softmax}(\cdot))$ layer on the $\mathbf{q}^k$ after 20 epochs. Each model is trained separately for each task.

### A.5   Natural Language Inference

Our baseline model/setup is essentially the same as that of Parikh et al. (2016). Let $[x_1, \ldots, x_n], [y_1, \ldots, y_m]$ be the premise/hypothesis, with the corresponding input representations $[\mathbf{x}_1, \ldots, \mathbf{x}_n], [\mathbf{y}_1, \ldots, \mathbf{y}_m]$. The input representations are obtained by a linear transformation of the 300-dimensional pretrained GloVe embeddings (Pennington et al., 2014) after normalizing the GloVe embeddings to have unit norm.[10] The pretrained embeddings remain fixed but the linear layer

---

[10]We use the GloVe embeddings pretrained over the 840 billion word Common Crawl, publicly available at `http://nlp.stanford.edu/projects/glove/`

(which is also 300-dimensional) is trained. Words not in the pretrained vocabulary are hashed to one of 100 Gaussian embeddings with mean 0 and standard deviation 1.

We concatenate each input representation with a convex combination of the other sentence's input representations (essentially performing *inter-sentence* attention), where the weights are determined through a dot product followed by a softmax,

$$
e_{ij} = f(\mathbf{x}_i)^\top f(\mathbf{y}_j) \quad \bar{\mathbf{x}}_i = \left[ \mathbf{x}_i; \sum_{j=1}^m \frac{\exp e_{ij}}{\sum_{k=1}^m \exp e_{ik}} \mathbf{y}_j \right] \quad \bar{\mathbf{y}}_j = \left[ \mathbf{y}_j; \sum_{i=1}^n \frac{\exp e_{ij}}{\sum_{k=1}^n \exp e_{kj}} \mathbf{x}_i \right]
$$

Here $f(\cdot)$ is an MLP. The new representations are fed through another MLP $g(\cdot)$, summed, combined with the final MLP $h(\cdot)$ and fed through a softmax layer to obtain a distribution over the labels $l$,

$$
\bar{\mathbf{x}} = \sum_{i=1}^n g(\bar{\mathbf{x}}_i) \qquad\qquad \bar{\mathbf{y}} = \sum_{j=1}^m g(\bar{\mathbf{y}}_j)
$$

$$
p(l \,|\, x_1, \ldots, x_n, y_1, \ldots, y_m) = \mathrm{softmax}(\mathbf{V}h([\bar{\mathbf{x}}; \bar{\mathbf{y}}]) + \mathbf{b})
$$

All the MLPs have 2-layers, 300 ReLU units, and dropout probability of 0.2. For structured/simple models, we first employ the bidirectional parsing LSTM (see A.1) to obtain the scores $\theta_{ij}$. In the structured case each word representation is simply concatenated with its soft-parent

$$
\hat{\mathbf{x}}_i = \left[ \mathbf{x}_i; \sum_{k=1}^n p(z_{ki} = 1 \,|\, x) \mathbf{x}_k \right]
$$

and $\hat{\mathbf{x}}_i$ (and analogously $\hat{\mathbf{y}}_j$) is used as the input to the above model. In the simple case (which closely corresponds to the *intra-sentence* attention model of Parikh et al. (2016)), we have

$$
\hat{\mathbf{x}}_i = \left[ \mathbf{x}_i; \sum_{k=1}^n \frac{\exp \theta_{ki}}{\sum_{l=1}^n \exp \theta_{li}} \mathbf{x}_k \right]
$$

The word embeddings for the parsing LSTMs are also initialized with GloVe, and the parsing layer is shared between the two sentences. The forward/backward LSTMs for the parsing layer are 100-dimensional.

Additional training details include: batch size of 32; training for 100 epochs with Adagrad (Duchi et al., 2011) where the global learning rate is 0.05 and sum of gradient squared is initialized to 0.1; parameter intialization over a Gaussian distribution with mean 0 and standard deviation 0.01; gradient normalization at 5. In the pretrained scenario, pretraining is done with Adam (Kingma & Ba, 2015) with learning rate equal to 0.01, and $\beta_1 = 0.9$, $\beta_2 = 0.999$.

## B    FORWARD/BACKWARD THROUGH THE INSIDE-OUTSIDE ALGORITHM

Figure 6 shows the procedure for obtaining the parsing marginals from the input potentials. This corresponds to running the inside-outside version of Eisner's algorithm (Eisner, 1996). The intermediate data structures used during the dynamic programming algorithm are the (log) inside tables $\alpha$, and the (log) outside tables $\beta$. Both $\alpha, \beta$ are of size $n \times n \times 2 \times 2$, where $n$ is the sentence length. First two dimensions encode the start/end index of the span (i.e. subtree). The third dimension encodes whether the root of the subtree is the left ($L$) or right ($R$) index of the span. The fourth dimension indicates if the span is complete (1) or incomplete (0). We can calculate the marginal distribution of each word's parent (for all words) in $O(n^3)$ using this algorithm.

Backward pass through the inside-outside algorithm is slightly more involved, but still takes $O(n^3)$ time. Figure 7 illustrates the backward procedure, which receives the gradient of the loss $\mathcal{L}$ with respect to the marginals, $\nabla_p^{\mathcal{L}}$, and computes the gradient of the loss with respect to the potentials $\nabla_\theta^{\mathcal{L}}$. The computations must be performed in the signed log-space semifield to handle log of negative values. See section 3.3 and Table 1 for more details.

**procedure** INSIDEOUTSIDE($\theta$)
 $\alpha, \beta \leftarrow -\infty$ ▷ Initialize log of inside ($\alpha$), outside ($\beta$) tables
 **for** $i = 1, \ldots, n$ **do**
 $\alpha[i, i, L, 1] \leftarrow 0$
 $\alpha[i, i, R, 1] \leftarrow 0$
 $\beta[1, n, R, 1] \leftarrow 0$
 **for** $k = 1, \ldots, n$ **do** ▷ Inside step
 **for** $s = 1, \ldots, n - k$ **do**
 $t \leftarrow s + k$
 $\alpha[s, t, R, 0] \leftarrow \bigoplus_{u \in [s, t-1]} \alpha[s, u, R, 1] \otimes \alpha[u+1, t, L, 1] \otimes \theta_{st}$
 $\alpha[s, t, L, 0] \leftarrow \bigoplus_{u \in [s, t-1]} \alpha[s, u, R, 1] \otimes \alpha[u+1, t, L, 1] \otimes \theta_{ts}$
 $\alpha[s, t, R, 1] \leftarrow \bigoplus_{u \in [s+1, t]} \alpha[s, u, R, 0] \otimes \alpha[u, t, R, 1]$
 $\alpha[s, t, L, 1] \leftarrow \bigoplus_{u \in [s, t-1]} \alpha[s, u, L, 1] \otimes \alpha[u, t, L, 0]$
 **for** $k = n, \ldots, 1$ **do** ▷ Outside step
 **for** $s = 1, \ldots, n - k$ **do**
 $t \leftarrow s + k$
 **for** $u = s + 1, \ldots, t$ **do**
 $\beta[s, u, R, 0] \leftarrow_\oplus \beta[s, t, R, 1] \otimes \alpha[u, t, R, 1]$
 $\beta[u, t, R, 1] \leftarrow_\oplus \beta[s, t, R, 1] \otimes \alpha[s, u, R, 0]$
 **if** $s > 1$ **then**
 **for** $u = s, \ldots, t - 1$ **do**
 $\beta[s, u, L, 1] \leftarrow_\oplus \beta[s, t, L, 1] \otimes \alpha[u, t, L, 0]$
 $\beta[u, t, L, 0] \leftarrow_\oplus \beta[s, t, L, 1] \otimes \alpha[s, u, L, 1]$
 **for** $u = s, \ldots, t - 1$ **do**
 $\beta[s, u, R, 1] \leftarrow_\oplus \beta[s, t, R, 0] \otimes \alpha[u+1, t, L, 1] \otimes \theta_{st}$
 $\beta[u+1, t, L, 1] \leftarrow_\oplus \beta[s, t, R, 0] \otimes \alpha[s, u, R, 1] \otimes \theta_{st}$
 **if** $s > 1$ **then**
 **for** $u = s, \ldots, t - 1$ **do**
 $\beta[s, u, R, 1] \leftarrow_\oplus \beta[s, t, L, 0] \otimes \alpha[u+1, t, L, 1] \otimes \theta_{ts}$
 $\beta[u+1, t, L, 1] \leftarrow_\oplus \beta[s, t, L, 0] \otimes \alpha[s, u, R, 1] \otimes \theta_{ts}$
 $A \leftarrow \alpha[1, n, R, 1]$ ▷ Log partition
 **for** $s = 1, \ldots, n - 1$ **do** ▷ Compute marginals. Note that $p[s, t] = p(z_{st} = 1 \mid x)$
 **for** $t = s + 1, \ldots, n$ **do**
 $p[s, t] \leftarrow \exp(\alpha[s, t, R, 0] \otimes \beta[s, t, R, 0] \otimes -A)$
 **if** $s > 1$ **then**
 $p[t, s] \leftarrow \exp(\alpha[s, t, L, 0] \otimes \beta[s, t, L, 0] \otimes -A)$
 **return** $p$

**Figure 6:** Forward step of the syntatic attention layer to compute the marginals, using the inside-outside algorithm (Baker, 1979) on the data structures of Eisner (1996). We assume the special root symbol is the first element of the sequence, and that the sentence length is $n$. Calculations are performed in log-space semifield with $\oplus = $ logadd and $\otimes = +$ for numerical precision. $a, b \leftarrow c$ means $a \leftarrow c$ and $b \leftarrow c$. $a \leftarrow_\oplus b$ means $a \leftarrow a \oplus b$.

**procedure** BACKPROPINSIDEOUTSIDE$(\theta, p, \nabla_p^{\mathcal{L}})$
 **for** $s, t = 1, \ldots, n; s \neq t$ **do** $\triangleright$ Backpropagation uses the identity $\nabla_\theta^{\mathcal{L}} = (p \odot \nabla_p^{\mathcal{L}})\nabla_\theta^{\log p}$
 $\delta[s,t] \leftarrow \log p[s,t] \otimes \log \nabla_p^{\mathcal{L}}[s,t]$ $\triangleright \delta = \log(p \odot \nabla_p^{\mathcal{L}})$
 $\nabla_\alpha^{\mathcal{L}}, \nabla_\beta^{\mathcal{L}}, \log \nabla_\theta^{\mathcal{L}} \leftarrow -\infty$ $\triangleright$ Initialize inside $(\nabla_\alpha^{\mathcal{L}})$, outside $(\nabla_\beta^{\mathcal{L}})$ gradients, and log of $\nabla_\theta^{\mathcal{L}}$
 **for** $s = 1, \ldots, n-1$ **do** $\triangleright$ Backpropagate $\delta$ to $\nabla_\alpha^{\mathcal{L}}$ and $\nabla_\beta^{\mathcal{L}}$
 **for** $t = s+1, \ldots, n$ **do**
 $\nabla_\alpha^{\mathcal{L}}[s,t,R,0], \nabla_\beta^{\mathcal{L}}[s,t,R,0] \leftarrow \delta[s,t]$
 $\nabla_\alpha^{\mathcal{L}}[1,n,R,1] \leftarrow_\oplus -\delta[s,t]$
 **if** $s > 1$ **then**
 $\nabla_\alpha^{\mathcal{L}}[s,t,L,0], \nabla_\beta^{\mathcal{L}}[s,t,L,0] \leftarrow \delta[t,s]$
 $\nabla_\alpha^{\mathcal{L}}[1,n,R,1] \leftarrow_\oplus -\delta[s,t]$
 **for** $k = 1, \ldots, n$ **do** $\triangleright$ Backpropagate through outside step
 **for** $s = 1, \ldots, n-k$ **do**
 $t \leftarrow s + k$
 $\nu \leftarrow \nabla_\beta^{\mathcal{L}}[s,t,R,0] \otimes \beta[s,t,R,0]$ $\triangleright \nu, \gamma$ are temporary values
 **for** $u = t, \ldots, n$ **do**
 $\nabla_\beta^{\mathcal{L}}[s,u,R,1], \nabla_\alpha^{\mathcal{L}}[t,u,R,1] \leftarrow_\oplus \nu \otimes \beta[s,u,R,1] \otimes \alpha[t,u,R,1]$
 **if** $s > 1$ **then**
 $\nu \leftarrow \nabla_\beta^{\mathcal{L}}[s,t,L,0] \otimes \beta[s,t,L,0]$
 **for** $u = 1, \ldots, s$ **do**
 $\nabla_\beta^{\mathcal{L}}[u,t,L,1], \nabla_\alpha^{\mathcal{L}}[u,s,L,1] \leftarrow_\oplus \nu \otimes \beta[u,t,L,1] \otimes \alpha[u,s,L,1]$
 $\nu \leftarrow \nabla_\beta^{\mathcal{L}}[s,t,L,1] \otimes \beta[s,t,L,1]$
 **for** $u = t, \ldots, n$ **do**
 $\nabla_\beta^{\mathcal{L}}[s,u,L,1], \nabla_\alpha^{\mathcal{L}}[t,u,L,0] \leftarrow_\oplus \nu \otimes \beta[s,u,L,1] \otimes \alpha[t,u,L,1]$
 **for** $u = 1, \ldots, s-1$ **do**
 $\gamma \leftarrow \beta[u,t,R,0] \otimes \alpha[u,s-1,R,1] \otimes \theta_{ut}$
 $\nabla_\beta^{\mathcal{L}}[u,t,R,0], \nabla_\alpha^{\mathcal{L}}[u,s-1,R,1], \log \nabla_\theta^{\mathcal{L}}[u,t] \leftarrow_\oplus \nu \otimes \gamma$
 $\gamma \leftarrow \beta[u,t,L,0] \otimes \alpha[u,s-1,R,1] \otimes \theta_{tu}$
 $\nabla_\beta^{\mathcal{L}}[u,t,L,0], \nabla_\alpha^{\mathcal{L}}[u,s-1,R,1], \log \nabla_\theta^{\mathcal{L}}[t,u] \leftarrow_\oplus \nu \otimes \gamma$
 $\nu \leftarrow \nabla_\beta^{\mathcal{L}}[s,t,R,1] \otimes \beta[s,t,R,1]$
 **for** $u = 1, \ldots, s$ **do**
 $\nabla_\beta^{\mathcal{L}}[u,t,R,1], \nabla_\alpha^{\mathcal{L}}[u,s,R,0] \leftarrow_\oplus \nu \otimes \beta[u,t,R,1] \otimes \alpha[u,s,R,0]$
 **for** $u = t+1, \ldots, n$ **do**
 $\gamma \leftarrow \beta[s,u,R,0] \otimes \alpha[t+1,u,L,1] \otimes \theta_{su}$
 $\nabla_\beta^{\mathcal{L}}[s,u,R,0], \nabla_\alpha^{\mathcal{L}}[t+1,u,L,1], \log \nabla_\theta^{\mathcal{L}}[s,u] \leftarrow_\oplus \nu \otimes \gamma$
 $\gamma \leftarrow \beta[s,u,L,0] \otimes \alpha[t+1,u,L,1] \otimes \theta_{us}$
 $\nabla_\beta^{\mathcal{L}}[s,u,L,0], \nabla_\alpha^{\mathcal{L}}[t+1,u,L,1], \log \nabla_\theta^{\mathcal{L}}[u,s] \leftarrow_\oplus \nu \otimes \gamma$
 **for** $k = n, \ldots, 1$ **do** $\triangleright$ Backpropagate through inside step
 **for** $s = 1, \ldots, n-k$ **do**
 $t \leftarrow s + k$
 $\nu \leftarrow \nabla_\alpha^{\mathcal{L}}[s,t,R,1] \otimes \alpha[s,t,R,1]$
 **for** $u = s+1, \ldots, t$ **do**
 $\nabla_\alpha^{\mathcal{L}}[u,t,R,0], \nabla_\alpha^{\mathcal{L}}[u,t,R,1] \leftarrow_\oplus \nu \otimes \alpha[s,u,R,0] \otimes \alpha[u,t,R,1]$
 **if** $s > 1$ **then**
 $\nu \leftarrow \nabla_\alpha^{\mathcal{L}}[s,t,L,1] \otimes \alpha[s,t,L,1]$
 **for** $u = s, \ldots, t-1$ **do**
 $\nabla_\alpha^{\mathcal{L}}[s,u,L,1], \nabla_\alpha^{\mathcal{L}}[u,t,L,0] \leftarrow_\oplus \nu \otimes \alpha[s,u,L,1] \otimes \alpha[u,t,L,0]$
 $\nu \leftarrow \nabla_\alpha^{\mathcal{L}}[s,t,L,0] \otimes \alpha[s,t,L,0]$
 **for** $u = s, \ldots, t-1$ **do**
 $\gamma \leftarrow \alpha[s,u,R,1] \otimes \alpha[u+1,t,L,1] \otimes \theta_{ts}$
 $\nabla_\alpha^{\mathcal{L}}[s,u,R,1], \nabla_\alpha^{\mathcal{L}}[u+1,t,L,1], \log \nabla_\theta^{\mathcal{L}}[t,s] \leftarrow_\oplus \nu \otimes \gamma$
 $\nu \leftarrow \nabla_\alpha^{\mathcal{L}}[s,t,R,0] \otimes \alpha[s,t,R,0]$
 **for** $u = s, \ldots, t-1$ **do**
 $\gamma \leftarrow \alpha[s,u,R,1] \otimes \alpha[u+1,t,L,1] \otimes \theta_{st}$
 $\nabla_\alpha^{\mathcal{L}}[s,u,R,1], \nabla_\alpha^{\mathcal{L}}[u+1,t,L,1], \log \nabla_\theta^{\mathcal{L}}[s,t] \leftarrow_\oplus \nu \otimes \gamma$
 **return** signexp $\log \nabla_\theta^{\mathcal{L}}$ $\triangleright$ Exponentiate log gradient, multiply by sign, and return $\nabla_\theta^{\mathcal{L}}$

**Figure 7:** Backpropagation through the inside-outside algorithm to calculate the gradient with respect to the input potentials. $\nabla_b^a$ denotes the Jacobian of $a$ with respect to $b$ (so $\nabla_\theta^{\mathcal{L}}$ is the gradient with respect to $\theta$). $a, b \leftarrow_\oplus c$ means $a \leftarrow a \oplus c$ and $b \leftarrow b \oplus c$.

