# Peer review of "Structured Attention Networks"

_ICLR 2017 — accepted_

[Official Review · AnonReviewer3 · rating 8 · confidence 3 · 16 Dec 2016]
**Interesting, clearly-written paper which proposes to extend attention over latent structures. However, experimental results on two real-world tasks only show small improvements over the baseline "simple" attention system.**
soundness 2 · clarity 3 · impact 2 · meaningful comparison 2

The authors propose to extend the “standard” attention mechanism, by extending it to consider a distribution over latent structures (e.g., alignments, syntactic parse trees, etc.). These latent variables are modeled as a graphical model with potentials derived from a neural network.

The paper is well-written and clear to understand. The proposed methods are evaluated on various problems, and in each case the “structured attention” models outperform baseline models (either one without attention, or using simple attention). For the two real-world tasks, the improvements obtained from the proposed approach are relatively small compared to the “simple” attention models, but the techniques are nonetheless interesting.

Main comments:
1. In the Japanese-English Machine Translation example, the relative difference in performance between the Sigmoid attention model, and the Structured attention model appears to be relatively small. In this case, I’m curious if the authors analyzed the attention alignments to determine whether the structured models resulted in better alignments. In other words, if ground-truth alignments are available for the dataset, or if they can be human-annotated for some test examples, it would be interesting to measure the quality of the alignments in addition to the BLEU metric.
2. In the final experiment on natural language inference, I thought it was a bit surprising that using pretrained syntactic attention layers did not appear to improve model performance, but instead appear to degrade performance. I was curious if the authors have any hypotheses for why this is the case?

Minor comments:
1. Typographical error: Equation 1: “p(z | x, q” → “p(z | x, q)”
2. Section 3.3: “Past work has demonstrated that the techniques necessary for this approach, … ” →  “Past work has demonstrated the techniques necessary for this approach, … ”

[Official Review · AnonReviewer2 · rating 8 · confidence 4 · 17 Dec 2016]
originality 2

This is a very nice paper. The writing of the paper is clear. It starts from the traditional attention mechanism case. By interpreting the attention variable z as a distribution conditioned on the input x and query q, the proposed method naturally treat them as latent variables in graphical models. The potentials are computed using the neural network.

Under this view, the paper shows traditional dependencies between variables (i.e. structures) can be modeled explicitly into attentions. This enables the use of classical graphical models such as CRF and semi-markov CRF in the attention mechanism to capture the dependencies naturally inherit in the linguistic structures.

The experiments of the paper prove the usefulness of the model in various level — seq2seq and tree structure etc. I think it’s solid and the experiments are carefully done. It also includes careful engineering such as normalizing the marginals in the model.

In sum, I think this is a solid contribution and the approach will benefit the research in other problems.

[Official Review · AnonReviewer1 · rating 8 · confidence 5 · 17 Dec 2016]
**solid paper**
soundness 2 · impact 1 · substance 1 · meaningful comparison 2 · recommendation (unofficial) 1

This is a solid paper that proposes to endow attention mechanisms with structure (the attention posterior probabilities becoming structured latent variables). Experiments are shown with segmental atention (as in semi-Markov models) and syntactic attention (as in projective dependency parsing), both in a synthetic task (tree transduction) and real world tasks (neural machine translation and natural language inference). There is a small gain in using structured attention over simple attention in the latter tasks. A clear accept.

The paper is very clear, the approach is novel and interesting, and the experiments seem to give a good proof of concept. However, the use of structured attention in neural MT seems doesn't seem to be fully exploited here: segmental attention could be a way of approaching neural phrase-based MT, and syntactic attention offers a way of incorporating latent syntax in MT -- these seem very promising directions. In particular it would be interesting to try to add some (semi-)supervision on these attention mechanisms (e.g. posterior marginals computed by an external parser) to see if that helps learning the attention components of the network, or at least help initializing them. 

This seems to be the first interesting use of the backprop of forward-backward/inside-outside (Stoyanov et al. 2011). As stated in sec 3.3., for general probabilistic models the forward step over structured attention corresponds to the computation of first-order moments (posterior marginals) while the backprop step corresponds to second-order moments (gradients of marginals wrt log-potentials, i.e., Hessian of log-partition function). This extends the applicability of the proposed approach to arbitrary graphical models where these quantities can be computed efficiently. E.g. is there a generalized matrix-tree formula that allows to do backprop for non-projective syntax? On the negative side, I suspect the need for second-order statistics may bring some numerical instability in some problems, caused by the use of the signed log-space field. Was this seen in practice?

Minor comments/typos:
- last paragraph of sec 1: "standard attention attention"
- third paragraph of sec 3.2: "the on log-potentials"
- sec 4.1, Results: "... as it has no information about the source ordering" -- what do you mean here?

[Final Decision · Program Chairs · 06 Feb 2017]
**ICLR committee final decision**

The area chair shares the reviewer's opinion and thinks that this is a very solid paper that deserves to be presented at ICLR. The idea is novel, well described and backed up by solid experiments (that show some empirical gains).